# Study of the Behavioral Characteristics of *Lutjanus erythropterus* on the Basis of the Distance and Structure of Artificial Reefs

**DOI:** 10.3390/biology14060644

**Published:** 2025-06-02

**Authors:** Manju Jiang, Yu Guo, Chuanxin Qin, Jia Zhang, Ankai Zhang

**Affiliations:** 1South China Sea Fisheries Research Institute, Chinese Academy of Fishery Sciences, Guangzhou 510380, China; jiangmj2022@163.com (M.J.); guoyu25895177@163.com (Y.G.); zj19922812035@163.com (J.Z.); 18925120768@163.com (A.Z.); 2Key Laboratory of Efficient Utilization and Processing of Marine Fishery Resources of Hainan Province, Sanya Tropical Fisheries Research Institute, Sanya 572018, China; 3College of Marine Sciences, Shanghai Ocean University, Shanghai 201306, China

**Keywords:** artificial reef, *Lutjanus erythropterus*, independent role, reef preference

## Abstract

With the further study of marine ecological protection and sustainable fishery development, artificial reefs, an effective means of marine ecological engineering, have gradually become the focus of research. Artificial reefs can not only provide marine organisms with habitats and breeding and foraging sites but also help promote the recovery and stabilization of marine ecosystems and increase their production. In this study, *Lutjanus erythropterus* was used as a target to analyze aggregation behavior comprehensively under different reef spacings and behavioral responses in the presence of many different types of reefs simultaneously. This study helps to understand the ecological needs of juvenile *L. erythropterus* more accurately, provides a scientific basis for optimizing the design and layout of artificial reefs, has a positive impact on the protection and proliferation of reef fish resources, maintains the balance of the marine ecosystem, and promotes the sustainable development of marine fisheries.

## 1. Introduction

Artificial fish reefs are human-made structures that are constructed and deployed in water areas. By changing the flow structure, these reefs can improve the ecological environment of coastal waters. Moreover, these reefs attract fish, thereby increasing the richness of fishery resources [1,2]. Compared with natural fish reefs, artificial fish reefs are largely more effective at increasing the diversity of fishery resources and improving fish catch [3,4,5]. To date, foreign researchers have focused mainly on the distribution and aggregation of fish near artificial fish reefs. Devices such as underwater cameras and echo sounders are usually used to analyze the behavioral patterns of fish [6,7,8]. In contrast, domestic researchers have focused more on the influences of the design and structure of artificial fish reefs on attraction effects [9,10]. The attraction effect generated after the deployment of artificial fish reefs is an extremely important reference index in the design of artificial fish reefs. When designing artificial fish reefs, multiple factors need to be comprehensively considered, including the shape of the reef body; the size, quantity, and shape of the openings; and the angle of the flow deflectors [11,12]. These factors affect the environment surrounding artificial fish reefs and thus influence the attraction effect. However, when artificial fish reefs are constructed for fish habitat restoration, two scientific issues need to be explored: (1) the integrity of the ecological service functions and the structure of the fish reef construction area, and (2) the adaptability and selectivity of fish behavior in the fish reef construction area. Exploring the behavioral characteristics of reef fish in artificial fish reef environments is helpful for revealing the internal principles of the fish aggregation phenomenon and further improving the ecological restoration effect of artificial fish reefs, ultimately enhancing fishery resources.

Reef-dwelling fish are species that mainly inhabit coral reefs, rocky reefs, and other reef areas and their surrounding regions. Reef-dwelling fish and their habitats are interdependent. The reefs provide the fish with foraging grounds, breeding sites, and safe shelters to avoid natural enemies. Moreover, the material circulation and energy flow of the reef ecosystem cannot do without the participation of the activities of reef-dwelling fish [13,14]. Artificial fish reefs are important artificial habitats that can improve the marine environment, simulate natural reefs, attract the aggregation of reef-dwelling fish, and create a suitable living space, which is highly important for maintaining population size. Exploring the interaction relationships between artificial fish reefs and fish is highly important for maintaining the stability of marine ecosystems [13].

*Lutjanus erythropterus,* a representative rocky reef fish, is a common tropical and subtropical fish species. These fish are mainly distributed in the South China Sea, the waters around Taiwan, and the coastal areas of Guangxi, Guangdong, and other regions in China [15]. In recent years, owing to the impacts of overfishing and extreme climate events, aquatic ecosystems have been damaged and the wild resources of the crimson snapper have sharply decreased, posing a serious threat to fishery resources and the ecological balance. Therefore, it has become necessary to understand the behavioral characteristics of this species in its habitat [16].

In recent years, research on artificial fish reefs has emerged continuously, but there are still some deficiencies in certain key areas. In previous studies, scholars often either focused on a single type of artificial fish reef or were limited to relatively closely spaced fish reefs. Relatively little research has been conducted on the comprehensive comparison of various types of fish reefs and the behaviors of fish under a wide range of fish reef spacings. Moreover, the current understanding of the behavioral patterns and preference mechanisms of juvenile *L. erythropterus* in different reef environments is not comprehensive enough. A reasonable layout of fish reefs can maximize their ecological and economic benefits. Therefore, research on the preferences of juvenile crimson snappers for different types of artificial fish reefs and the interaction distances between fish reefs is particularly urgent and necessary.

Previous research on the behavior of juvenile *L. erythropterus* has shown that when the distance between fish reefs is greater than three times the size of a single reef, the reefs act as independent entities. However, when the distance is less than 0.5 times the size of a single reef, the reefs form an integrated structure and lose their ability to function as a group of independent fish reefs. Therefore, in this study, by observing the aggregation behaviors of crimson snappers under different fish reef spacings and their behavioral responses when there were two or four different types of fish reefs in the same group (room-shaped artificial reef, RAR; lobe-shaped artificial reef, LAR; cubic artificial reef, CAR; and stacked artificial reef, SAR), a comprehensive analysis of their preference patterns was performed. The aim of this study was to systematically explore the behavioral performance of juvenile crimson snappers in a diverse artificial fish reef environment. This study helps to more accurately understand the ecological needs of juvenile crimson snappers and provides a scientific basis for optimizing the design and layout of artificial fish reefs. The research results not only provide practical references for the selection, configuration, and spacing setting of artificial fish reefs in marine ranches but also positively impact the protection and enhancement of rocky reef fish resources, the maintenance of marine ecological balance, and the promotion of the sustainable development of marine fisheries.

## 2. Materials and Methods

### 2.1. Experimental Fish and Culture Conditions

The 500 *Lutjanus erythropterus* individuals used for this experiment were provided by the Tropical Aquatic Research and Development Center, South China Sea Fisheries Research Institute, Chinese Academy of Fisheries Sciences. These fish were undamaged, with a total length of 9.42 ± 1.09 cm, body length of 7.84 ± 1.08 cm, body height of 3.81 ± 0.24 cm, and body weight of 12.71 ± 2.68 g. The experiments were carried out from July to August 2023 at the Tropical Aquatic Research and Development Center of Lingshui Lizu Autonomous County, Hainan Province.

The experimental fish were temporarily housed in culture tanks with a yellow background and dimensions of 120 cm × 120 cm × 100 cm (L × W × H). The seawater used originated from the local sea area and was put into use after flow-through sand filtration. The water temperature was maintained at 31.26 ± 0.38 °C, the salinity was 28.0%, the pH was 7.98 ± 0.10, the light conditions were natural light, and the dissolved oxygen concentration was at or above 7 mg/L. The holding tank was continuously aerated for 24 h. The fish were fed twice a day, and the tank was cleaned by siphoning after each feeding for 0.5 h. The feeding of the fish was stopped 24 h before the start of the experiment, and the fish were acclimatized under the above conditions for two months.

### 2.2. Experimental Setup

The artificial reef model used in this experiment was a box reef, which was made at a 1:10 scale consistent with the common artificial reefs used in South China Sea marine farms. The model was composed of five identical acrylic panels with dimensions of 30 cm × 30 cm × 30 cm (L × W × H). In previous studies, the body height of juvenile *L. erythropterus* was 4 cm. On the basis of the thigmotactic characteristics of juvenile *L. erythropterus* toward fish reefs with different opening shapes [1], the opening shape of the fish reef in this experiment was set as circular, and its diameter was set to twice the body height of the juvenile fish (8.0 cm).

#### 2.2.1. Reef Selection for the Independent Effect Distance

To clarify whether the reefs had a combined effect or independent effect when 2 artificial reefs were present, 2 cubic reefs were selected, as shown in Figure 1c.

#### 2.2.2. Reef Selection for Behavioral Research

To investigate the reef type preferred by *L. erythropterus* when two or four different reefs were present at the same time, four types of artificial reef models, namely room-shaped artificial reef (RAR), lobe-shaped artificial reef (LAR), cubic artificial reef (CAR), and stacked artificial reef (SAR), were prepared, as shown in the images in Figure 1.

### 2.3. Experimental Reef Placement

#### 2.3.1. Determination of the Distance at Which Reefs Result in Independent Effects in Fish

The experimental tanks had dimensions of 230 cm × 230 cm × 100 cm, 320 cm × 230 cm × 100 cm, and 550 cm × 230 cm × 100 cm. These tanks had blue backgrounds and 70 cm water depths. Tanks with dimensions of 230 cm × 230 cm × 100 cm tanks were used at distances of 0.5 times the reef length (0.5 L), 1.0 times the reef length (1.0 L), and 1.5 times the reef length (1.5 L), and the placement area is shown in Figure 2a–c. Tanks with dimensions of 320 cm × 230 cm × 100 cm were used at distances of 2.0 times the reef length (2.0 L) and 2.5 times the reef length (2.5 L), and the placement area is shown in Figure 2d,e. Tanks with dimensions of 550 cm × 230 cm × 100 cm were used at distances of 3.0 times the reef length (3.0 L), and the placement area is shown in Figure 2f. The configuration of groups with different reef spacings is shown in Figure 3, with A, B, C, D, E, and F representing the average subareas of the distance between artificial reefs from far to near, respectively.

#### 2.3.2. Zoning of Tank

The experimental tank was a 550 cm × 230 cm × 100 cm rectangular tank with a blue background and a water depth of 70 cm. The experimental tank was divided into 10 equal parts, denoted I–X, and the partitioning pattern was based on the principle of equalizing each area according to the principle of average partitioning, as noted by Zhang Zonghang et al. [17]. The partitioning method is shown in Figure 4. Artificial reefs were placed in zones IV and VIII in the experiments to test the *L. erythropterus* reef preferences in the presence of two different artificial reef models. One artificial reef was placed at points A, B, C, and D in the experiments to test the *L. erythropterus* reef preferences in the presence of four different artificial reef models to ensure that *L. erythropterus* was equidistant from each of the artificial reefs.

### 2.4. Experimental Conditions

Environmental indicators, such as dissolved oxygen, salinity, water temperature, and pH, were monitored daily during the experimental period to maintain consistent conditions, and the light was natural light. The light intensity of each treatment group during the experimental period is shown in Figure 5.

### 2.5. Experimental Methods

#### 2.5.1. The Independent Effect Distance of Fish Reefs

For each experiment, 10 *L. erythropterus* individuals were randomly selected from their holding tanks. The fish were acclimatized in the tank for 30 min before the start of the experiment, and the reef spacing was set to 0.5, 1.0, 1.5, 2.0, 2.5, and 3.0 times. The reef models were placed at specific locations in the tanks. After the reef model was placed in the tank, we started recording video. The camera recording period was from 8:00 to 18:00. Overall monitoring of the experimental tank was carried out using two UVision IPC325L-IR3-UF high-definition cameras (20 fps, resolution 2592 × 1944) mounted directly above the experimental tank. Moreover, two underwater cameras of the same model, ONVIF HDIPC (30 fps, resolution 1920 × 1080), were fixed to a stainless steel bracket to monitor the behavior of juvenile *L. erythropterus*. The video from the experiment was transferred to a hard disk in real time. From the video, screenshots were taken every 30 min via the video recording device to record and analyze the distributions of fish in the tanks during the day. The experiment was repeated for each treatment group (3 L).

#### 2.5.2. Experiments on Two Types of Reef Models

The experimental equipment and video recording protocols were the same as those described in Section 2.5.1. For each experiment, 20 healthy individuals were randomly removed from their holding tanks. In addition to taking screenshots every 30 min from the video recordings, which were used to record and analyze the distributions of fish in the experiment during the daytime, 10 min of video data were collected every 30 min, and the residence time of the test fish was analyzed by selecting three individual fish for tracking.

Repeated experiments (3 L) were set up for each treatment group, and the positions of the two model reefs were adjusted each time. The positions of the experimental model reefs were changed to eliminate the effect of their position on the habitat selectivity of *L. erythropterus*.

#### 2.5.3. Experiments on Four Types of Reef Models

The experimental method was the same as that described in Section 2.5.2. The reefs were placed at points A, B, C, and D shown in the figure, and one artificial reef was placed in each area. The experiment was repeated (3 L), and the four model reefs were repositioned in a clockwise direction each time. The positions of the model reefs were changed to eliminate their effects on the habitat selectivity of *L. erythropterus*.

During the experiments described in Section 2.5.1, Section 2.5.2 and Section 2.5.3, there was no bait, the seawater flow was turned off, and the oxygen supply was stopped during the daytime and resumed at night. To prevent the experimental fish from acclimating to the tank environment and the artificial reef model, the experimental fish used were removed at the end of each day of the experimental period and placed in a separate culture tank for temporary rearing. All the water used was replaced at the end of each day to prevent chemical signals from being retained in the water. To ensure the accuracy of the results, fences were set up to minimize interference from the movement of the researchers.

### 2.6. Statistical and Analytical Methods

The distances of the reefs to retain the independent effects were determined by recording and calculating the average value of each area to measure the independence of the aggregation effect produced by the two reefs on juvenile *L. erythropterus*. The number of juvenile *L. erythropterus* in each area under each experimental group was calculated by the following formulae:$$0.5\;L:\;A=\bar{A_{1}+A_{2}}$$$$1.0\;L:A=\bar{A_{1}+A_{2}}\;\;\;B=\bar{B_{1}+B_{2}}$$$$1.5\;L:\;A=\bar{A_{1}+A_{2}}\;\;\;B=\bar{B_{1}+B_{2}}\;\;\;C=\bar{C_{1}+C_{2}}$$$$2.0\;L:\;A=\bar{A_{1}+A_{2}}\;\;\;B=\bar{B_{1}+B_{2}}\;\;\;C=\bar{C_{1}+C_{2}}\;\;\;D=\bar{D_{1}+D_{2}}$$$$2.5\;L:\;A=\bar{A_{1}+A_{2}}\;\;\;B=\bar{B_{1}+B_{2}}\;\;\;C=\bar{C_{1}+C_{2}}\;\;\;D=\bar{D_{1}+D_{2}}\;\;\;E=\bar{E_{1}+E_{2}}$$$$3.0\;L:\;A=\bar{A_{1}+A_{2}}\;\;\;B=\bar{B_{1}+B_{2}}\;\;\;C=\bar{C_{1}+C_{2}}\;\;\;D=\bar{D_{1}+D_{2}}\;\;\;E=\bar{E_{1}+E_{2}}\;\;\;F=\bar{F_{1}+F_{2}}$$

In this experiment, the reefing behavior of *L. erythropterus* was quantitatively measured by the mean distribution rate (MDR) of the fish in each area [17]. For the mean distribution rate (MDR) of the fish, the formula was as follows:$$MDR\left(\%\right)=\frac{1}{mn}\sum_{i=1}^{m}n_{i}\times 100\%$$
where $n_{i}$ is the number of ith observations of the distribution of experimental fish in a zone, m is the number of observations, and n is the total number of experimental fish.

In each experiment, we calculated the time from the start of the experiment to the time at which the first fish approached a reef model, interpreting this result as an indication of a fish’s willingness to venture out and explore unknown locations and potentially dangerous areas (i.e., exploratory), which was an indication of the fish’s level of courage. The response time was the time at which the first fish reached the reef.

We defined the time that the fish stayed on each artificial reef model within 10 min as the residence time. In the preliminary stage, we randomly selected one fish from each artificial reef every hour with the help of the video recording process, and we determined the time it stayed on the artificial reef. The longest time was 427 s; thus, the time range was determined to be 10 min.

The experimental data were statistically analyzed using SPSS 26.0, and the results are expressed as means ± standard deviation (x ± SD). The Kolmogorov-Smirnov test and Levene’s test were used to analyze normality and the chi-squared value of variance, respectively. The independent-sample t test was used to compare the differences in response times between an LAR and RAR, between a CAR and SAR, and between an LAR and CAR when they were placed together. The differences in the mean distribution rates of fish between a CAR and SAR and between an LAR and CAR were determined as well. Given the nonnormal distribution or variance heterogeneity, a Mann-Whitney U test was performed for the remaining data with both reef types. One-way ANOVA was used to compare the differences in the response times and the mean distribution rates of *L. erythropterus* in different reef types when RAR, LAR, CAR, and SAR were present together. One-way ANOVA revealed significant differences (*p* < 0.05) when the data were compared two by two with the least significant difference (LSD) post hoc test. Under nonnormal distribution or variance heterogeneity, Kruskal-Wallis H tests were conducted to determine the differences in the *L. erythropterus* residence times across different reef types in the presence of an RAR, LAR, CAR, or SAR. The resulting data were plotted via GraphPad 8.0.2 software. Differences were considered statistically significant at the probability level of *p* < 0.05.

## 3. Results

### 3.1. Behavioral Characteristics of Juvenile Lutjanus erythropterus

#### 3.1.1. Determination of the Independent Effect Distance of Fish Reefs

Video analysis revealed that at a 0.5 L reef length, juvenile *L. erythropterus* gathered in between the reefs and frequently shuttled between them, with 4–6 fish circling in this central area. The number of fish around the pool and at the four corners of the pool was low, with a short residence time in the reef. When the distance between the reefs was 1.0 L, 1.5 L, and 2.0 L, juvenile *L. erythropterus* still shuttled between the two reefs. However, their circling activity gradually decreased until it stopped. Moreover, the activity between the two areas decreased, the speed of fish movement significantly decreased, and the number of juvenile *L. erythropterus* around the periphery of the pools and at the corners of the pools increased. At 2.5 L and 3.0 L, the shuttle activities between the two reefs decreased and even stopped, the juvenile *L. erythropterus* began to move frequently along the other three sides of the reefs, the number of juveniles in the corners and around the tanks increased, and the speed of fish movement slowed. The distributions of juvenile *L. erythropterus* at different reef spacings are shown in Figure 6.

#### 3.1.2. Preference of *L. erythropterus* for Fish Reef

Before being placed in the reef model, owing to stress from fishing and changes in the environment, the juvenile *L. erythropterus* initially exhibited abnormal behaviors and immediately swam in all directions of the pool, with a high initial speed. After 10–15 min, the swimming speed of juvenile *L. erythropterus* decreased, and they tended to stick to the walls of the pool and often swam toward the middle, often chasing each other and speeding up. The fish swam in groups more frequently.

After being placed in the artificial reef model, *L. erythropterus* first fled in fear and swam in a direction away from the turbulent area and the artificial reef model. After a period of adaptation to stress, the swimming speed of the fish decreased, and the swimming speed was limited to the corners of the tank. The fish seldom swam together or gathered in one place. After approximately 2 min, some fish swam toward the artificial reef and stayed at the edge of the artificial reef model with their heads facing the model to observe the internal environment. Other fish swam slowly around the model for a week and then swam around the vicinity. The fish then entered the reef, and the frequency at which they entered and exited the model increased. When there were 2–3 fish inside the reef, other juvenile *L. erythropterus* were driven out when they tried to enter the model, and the juvenile

*L. erythropterus* that were in the reef were aggressive toward individuals on the outside, displaying some degree of territorial behavior. After adapting to the new environment, the fish began to become comfortable, travel to various areas, and stay inside the reef and in the surrounding and shaded areas. Furthermore, no significant rejection of the artificial reef was observed during the experiment. The distribution of juvenile L. erythropterus in the different treatment groups is shown in Figure 7.

### 3.2. Determination of the Distances of the Fish Reefs to Retain the Independent Effects

The abundance of juvenile *L. erythropterus* in all areas of the experimental tanks differed significantly (*p* < 0.05) at 2.5 L and 3.0 L. At 2.5 L, areas E and D were significantly more populated than areas A, B and C. At 3.0 L, areas E and F were significantly more populated than the other areas.

The number of juvenile *L. erythropterus* tended to decrease overall with increasing distance from the reef in each artificial reef spacing model, and the number of *L. erythropterus* began to decrease with increasing distance at 2.5 d and 3.0 d (Figure 8).

There was a significant difference in the number of juvenile *L. erythropterus* in Zone A under different reef spacing conditions, with the number of juvenile *L. erythropterus* in Zone A decreasing as the distance between the two reefs increased (Figure 9). The maximum and minimum values of the number of juvenile *L. erythropterus* in Zone A in each experimental group were 1.80 (0.5 L) and 0.04 (3.0 L), respectively. Juvenile *L. erythropterus* had the greatest activity in Zone A at 0.5L and the lowest activity at 3.0 L.

### 3.3. Behavioral Characteristics of the Two Types of Artificial Fish Reef Models

#### 3.3.1. Response Time

In terms of the response time of juvenile *L. erythropterus* in the presence of two different artificial reef models, the difference between the treatment groups was not significant (*p* > 0.05) (Figure 10). When RARs and LARs were present, the response times were 166.33 ± 45.65 s and 113.33 ± 43.29 s, respectively. The response times of juvenile *L. erythropterus* to LARs were shorter than those to RARs. The response times were 124.67 ± 23.80 s and 116.33 ± 26.35 s in the presence of CARs and SARs, respectively. Juvenile *L. erythropterus* had shorter response times to SARs than to CARs. The response times were 123.67 ± 44.00 s and 107.33 ± 36.20 s in the presence of both CARs and LARs, respectively, and juvenile *L. erythropterus* had shorter response times to LARs than to CARs.

#### 3.3.2. Mean Distribution Rate

There was a significant difference (*p* < 0.05) between the treatment groups in the mean distribution rate of juvenile *L. erythropterus* in the presence of two different artificial reef models (Figure 11), as evidenced by the fact that in the presence of RARs and LARs, the mean distribution rates were 13.58 ± 1.61% and 25.33 ± 1.88%, respectively. Moreover, juvenile *L. erythropterus* had a greater mean distribution rate in the LAR than in the RAR. When CARs and SARs were present, the mean distribution rates were 29.17 ± 2.98% and 16.08 ± 2.88%, respectively. In addition, the mean distribution rate of juvenile *L. erythropterus* was greater in the CAR than in the SAR. When both CARs and LARs were present, the mean distribution rates were 25.17 ± 3.17% and 15.00 ± 2.88%, respectively. The mean distribution rate of juvenile *L. erythropterus* was greater in the CAR than in the LAR.

#### 3.3.3. Residence Time

In terms of the residence time of juvenile *L. erythropterus* in the presence of two different artificial reef models, there was a significant difference (*p* < 0.05) between the RAR and LAR treatment groups and the CAR and SAR treatment groups (Figure 12), with 36.49 ± 38.59 s and 53.83 ± 34.75 s being the residence times of juvenile *L. erythropterus* in the RAR and LAR, respectively. The juvenile *L. erythropterus* had a longer residence time in the LAR. In the presence of the CAR and SAR, the residence times were 65.66 ± 59.38 s and 47.16 ± 34.78 s, respectively. The juvenile *L. erythropterus* had a longer residence time in the CAR than in the SAR. When the CAR and LAR were present, the residence times were 77.93 ± 70.40 s and 64.03 ± 57.63 s, respectively. The juvenile *L. erythropterus* had a longer residence time in the CAR than in the LAR.

### 3.4. Behavioral Characteristics of the Four Types of Artificial Fish Reef Models

#### 3.4.1. Response Time

In terms of the response times of juvenile *L. erythropterus* in different reefs in the presence of four different artificial reef models, the differences between the treatment groups were not significant (*p* > 0.05) (Figure 13a), and the response times of juvenile *L. erythropterus* to the RAR, LAR, CAR and SAR were 59.33 ± 9.61 s, 35.33 ± 3.51 s, 46.67 ± 16.80 s, and 41.33 ± 18.01 s, respectively. The response times were in the order LAR < SAR < CAR < RAR.

#### 3.4.2. Mean Distribution Rate

The mean distribution rates of juvenile *L. erythropterus* in different reefs in the presence of the four different artificial reef models were significantly different between the treatment groups (*p* < 0.05) (Figure 13b), with the mean distribution rates of juvenile *L. erythropterus* in the RAR, LAR, CAR, and SAR models being 8.17 ± 0.76%, 20.50 ± 5.91%, 27.42 ± 4.69%, and 12.75 ± 1.52%, respectively. The average distribution rates in the reef models were in the order CAR > LAR > SAR > RAR.

#### 3.4.3. Residence Time

Significant differences (*p* < 0.05) were detected between the treatment groups in terms of the residence time of juvenile *L. erythropterus* in different reefs in the presence of four different artificial reef models (Figure 13c), with the residence times of juvenile *L. erythropterus* in the RAR, LAR, CAR, and SAR being 41.34 ± 44.09 s, 66.62 ± 58.20 s, 94.12 ± 111.21 s, and 54.32 ± 64.15 s, respectively. The residence times in the CAR model were greater than those in the LAR model, and those in the LAR model were greater than those in the SAR model.

## 4. Discussion

### 4.1. Determination of the Independent Effect of Distance of Fish Reefs and Analysis of Ecological Effects

Artificial reefs, important parts of marine ranching projects, play key roles in protecting marine biodiversity and restoring coastal habitats. Granneman et al. [18] reported that compared with natural reefs, artificial reefs can increase fish density and biodiversity, and in their experiments, artificial reefs with different spacings had significant effects on attracting fish. In addition, reefs of different shapes and structures produce complex ecological effects, and according to previous experiments, *Lutjanus erythropterus* prefers cubic reefs [19]; therefore, a CAR was chosen as the experimental reef type in this study.

Scholars in countries other than China often use equipment such as underwater cameras and echosounders to analyze fish behavior near artificial reefs [6,20,21], whereas researchers in China have focused on the influences of the design and structure of artificial reefs on their trapping effects [22,23]. In recent years, many studies have shown that the topography, height, spacing, and opening ratio of artificial reefs affect their biotrapping effect and ecological and economic benefits [24,25], and Campbell et al. [26] and Liu et al. [27] noted that the arrangement and combination of artificial reefs have important impacts on the ecological value of the area in which they are located. Zhang et al. [28] further explored this effect by studying the layout of submarine artificial reefs under different spacing conditions. Yu et al. [29,30] investigated the hydrodynamic characteristics of submerged reefs and trapezoidal reefs at different spacing intervals. In this study, we explored the effect of the area between two artificial reefs on fish entrainment under six configurations through indoor flume simulation experiments, aiming to determine whether *L. erythropterus* displays a combined or independent entrainment effect on existing artificial reefs and to determine whether independent effects exist on the basis of the number of *L. erythropterus* entrained. The results of this study can serve as a reference for determining the entrainment effects of artificial reefs on other fish species.

A study by Mingjian Lin et al. [31] revealed that reef spacing affects the ecological effects of reefs. The present study consisted of a hydrostatic experiment, which needs to be combined with flow rate studies in the future to better reflect the conditions in the natural state. In addition, the horizontal distance gradient method was used in this study, which does not provide a detailed study of reef distance or accurately define the threshold for the independent action of *L. erythropterus*, which can be further explored in the future. The ecological effects of the long-term placement of artificial reefs need to be evaluated in future long-term experiments [32].

### 4.2. Analysis of the Preference of Juvenile Lutjanus erythropterus for Different Artificial Reef Models

Through shading, artificial reefs provide hiding spaces and habitats for fish, attracting many fish [23,33,34,35]. *L. erythropterus* is a warmwater demersal fish that prefers to live in low-light areas such as muddy seabeds and shallow rocky reefs [36,37]. The four types of artificial reefs selected for the present experiments tended to attract juvenile *L. erythropterus* due to the shade provided by the artificial reefs, especially under external stimulation, which resulted in a reduction in the occurrence rate of fish at the edges of the tanks, which is in line with most studies on rocky reef fish [23,34,38,39,40]. All four models of artificial reefs had significant effects on attracting juvenile *L. erythropterus*. These results can be used as a reference for resource conservation and enhancement.

Previous studies have shown that artificial reef models of different shapes and structures have different entrainment effects on juvenile *L. erythropterus* [41,42,43,44]. Jiong [45] indicated that time indicators are more intuitive in reflecting differences in fish domestication and behavior. In this experiment, we explored the relationship between the reef model and the behavior of *L. erythropterus* relative to the response time, average distribution rate, and residence time values. The results revealed that the response times of *L. erythropterus* to different reefs were different, and the response times were usually shorter for an LAR and longer for an RAR. It was inferred that the response time might be related to the light and shading effects inside the reefs and that as an RAR has lower light intensity and less effective space, the exploratory behaviors of *L. erythropterus* might be greater in the presence of an RAR than in the presence of an LAR, which has slightly greater light intensity and more space.

Partridge et al. [46] noted the aggregation behaviors of fish groups and individuals. In their study, *L. erythropterus* swam in pairs and clustered in the reef after the artificial reef was placed; however, the average distribution of fish in the reef area did not reach 100%, which may have occurred because the domain behavior is prevalent in rocky reef fish [47].

In this study, the behavioral responses of *L. erythropterus* were analyzed qualitatively and quantitatively in an artificial reef environment; however, the responses differed from those observed in actual sea areas [48,49,50]. In the future, we should combine diving observations, underwater camera monitoring, and remote sensing technology to observe the effects of artificial reefs placed in sea areas on fish aggregation. Moreover, we should combine these methods with indoor experiments to provide a basis for the selection of reef materials, the optimization of structural design and the determination of reef spacing.

## 5. Conclusions

In this study, juvenile *Lutjanus erythropterus* with a body height of 3.81 ± 0.24 cm were used. When the distance between the reefs was 0.5 L, the juvenile *L. erythropterus* moved most frequently between the reefs, and the trapping effect was significant. At 0.5 L, 1.0 L, 1.5 L and 2.0 L, there were no significant differences in the number of juveniles in each area. At 2.5 L and 3.0 L, there were significant differences in the number of juveniles in each area, and the number of juveniles tended to level off with increasing distance from the reef. It was inferred that the reefs had combined effects at reef distances of 0.5 L–2.0 L and had independent effects at reef distances of 2.5 L and 3.0 L. Juvenile *L. erythropterus* prefer cubic artificial reef models with circular opening shapes and an opening size twice their body height when there are two and four different artificial reef models present. For this type of reef, the reef-attracting tendency is the strongest, their staying time is the longest, and the attracting effect is the best.

## Figures and Tables

**Figure 1 biology-14-00644-f001:**
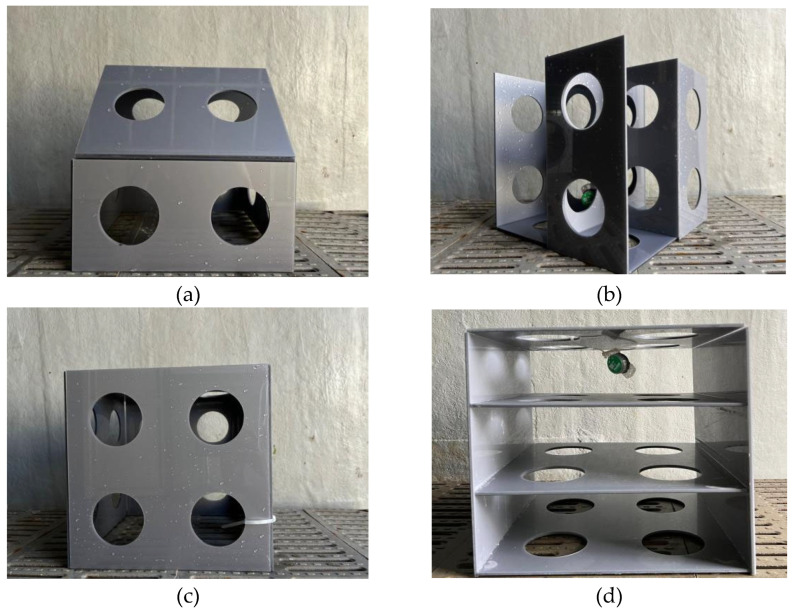
Images of the artificial reef model. (**a**) Room-shaped artificial reef. (**b**) Lobe-shaped artificial reef. (**c**) Cubic artificial reef. (**d**) Stacked artificial reef.

**Figure 2 biology-14-00644-f002:**
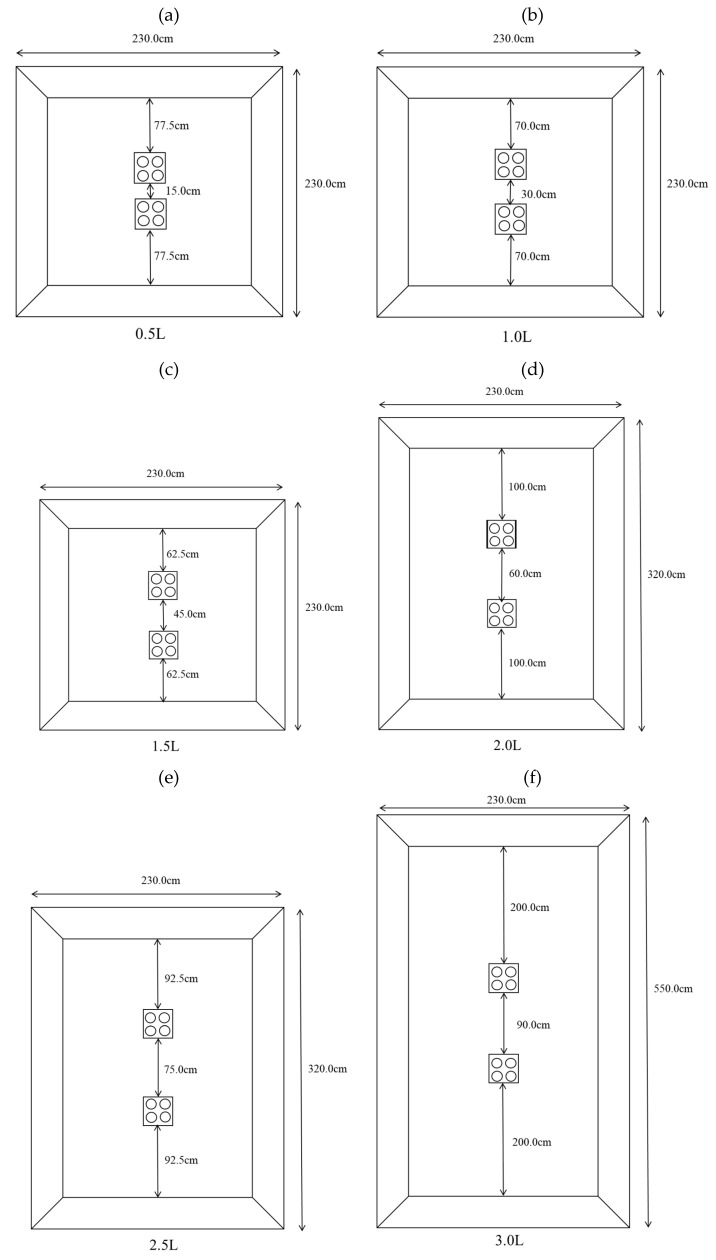
Schematics of the areas where the reefs were located. (**a**) 0.5 L; (**b**) 1.0 L; (**c**) 1.5 L; (**d**) 2.0 L; (**e**) 2.5 L; (**f**) 3.0 L.

**Figure 3 biology-14-00644-f003:**
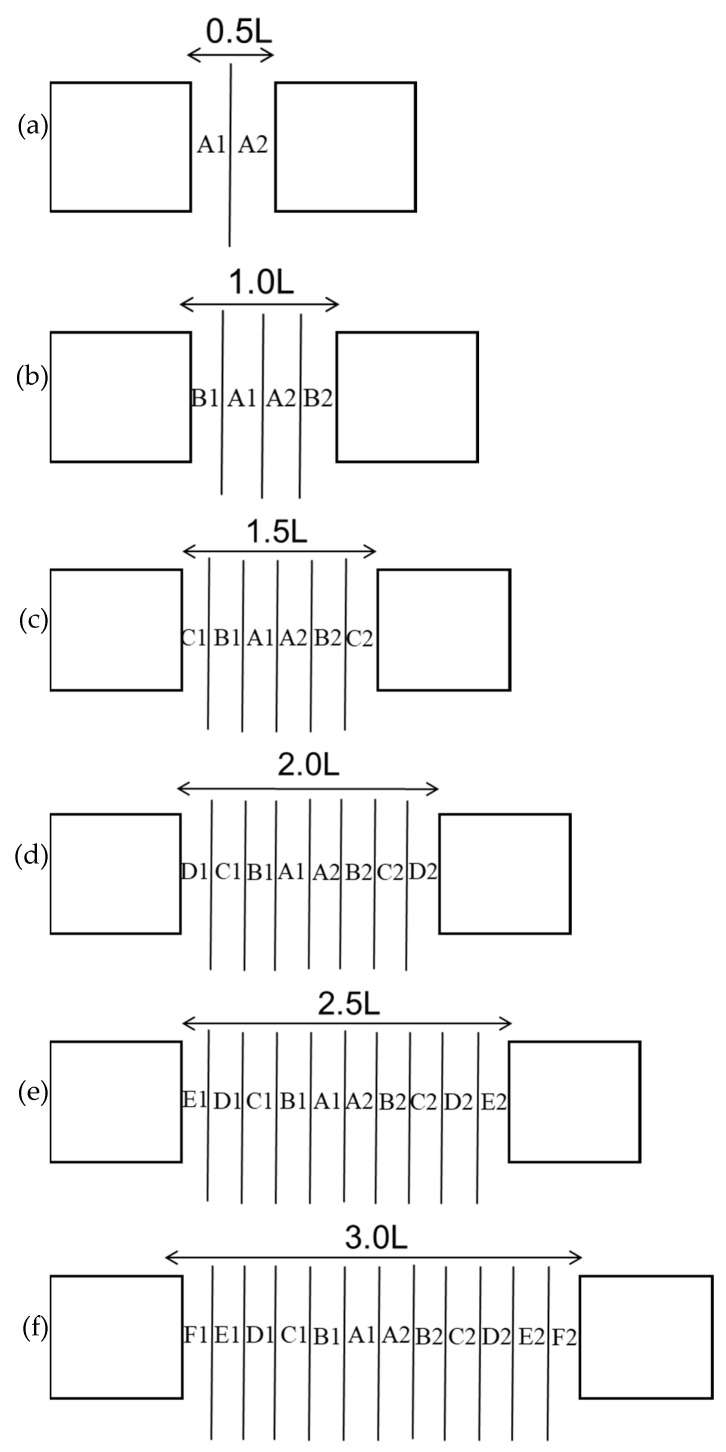
Maps of the interreef body zonations at different reef spacings. (**a**) 0.5 times the reef length; (**b**) 1.0 times the reef length; (**c**) 1.5 times the reef length; (**d**) 2.0 times the reef length; (**e**) 2.5 times the reef length; and (**f**) 3.0 times the reef length.

**Figure 4 biology-14-00644-f004:**
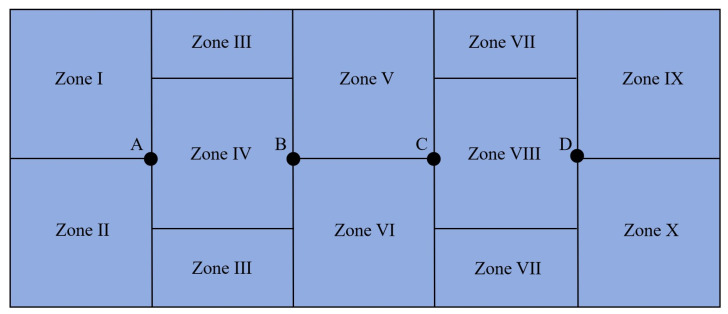
Schematic of the division of the bottom area of the test tank.

**Figure 5 biology-14-00644-f005:**
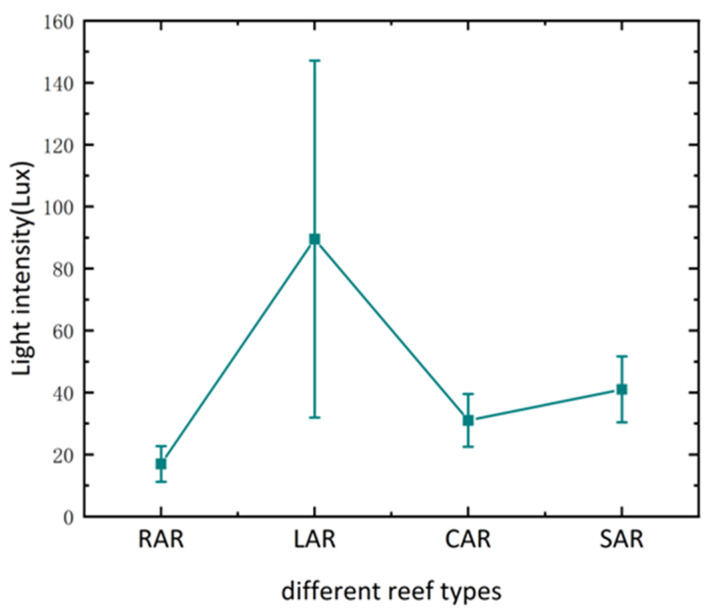
Light intensity within the reef in each artificial reef model.

**Figure 6 biology-14-00644-f006:**
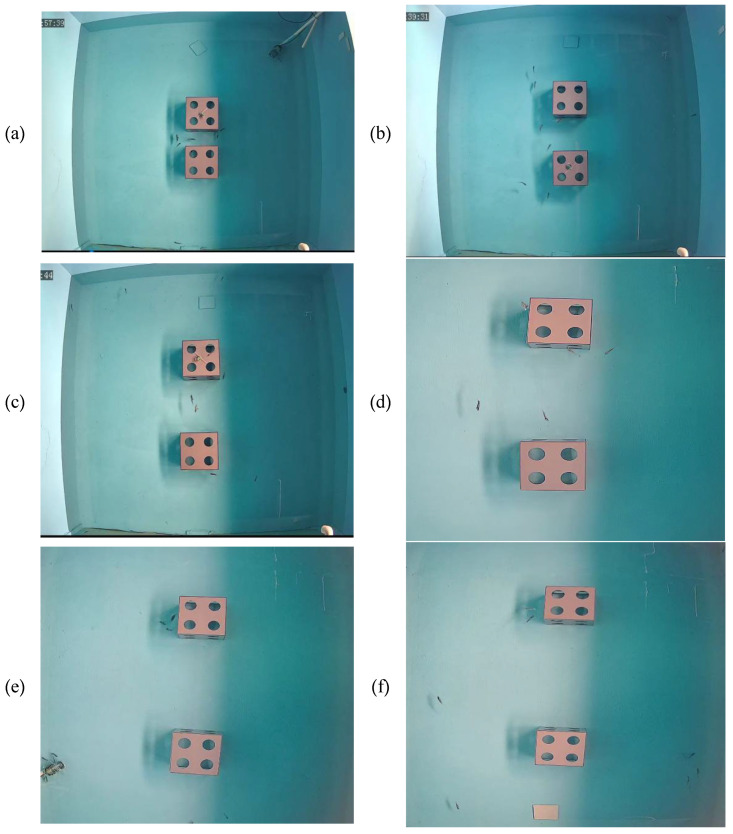
Distributions of juvenile *L. erythropterus* at different reef spacings: (**a**) 0.5 L; (**b**) 1.0 L; (**c**) 1.5 L; (**d**) 2.0 L; (**e**) 2.5 L; and (**f**) 3.0 L.

**Figure 7 biology-14-00644-f007:**
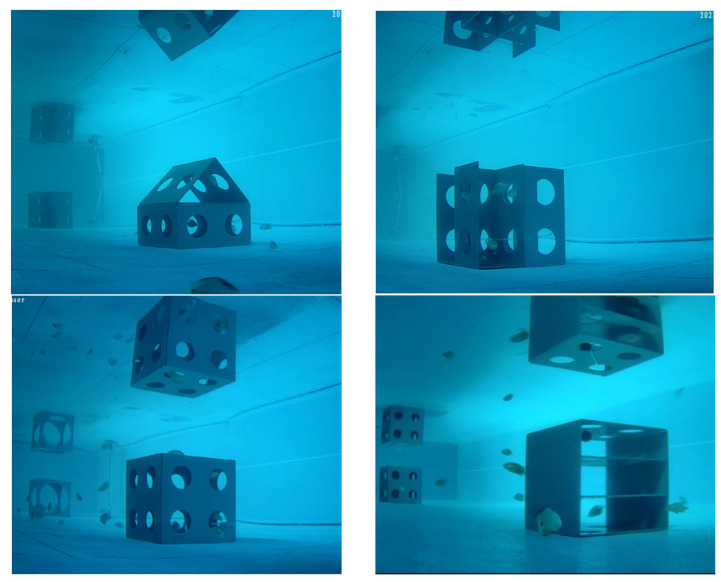
Distribution of juvenile *L. erythropterus* under different artificial reef assemblages.

**Figure 8 biology-14-00644-f008:**
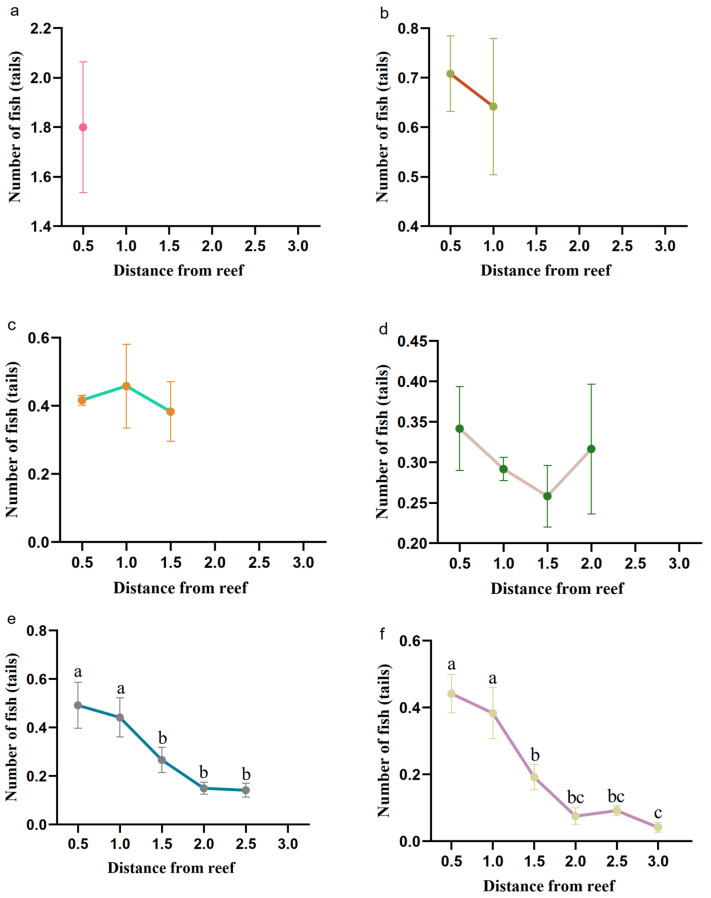
Number of fish in each area at different reef spacings. (**a**) 0.5 L; (**b**) 1.0 L; (**c**) 1.5 L; (**d**) 2.0 L; (**e**) 2.5 L; and (**f**) 3.0 L. Diﬀerent letters indicate the signiﬁcant diﬀerence at the level of 0.05.

**Figure 9 biology-14-00644-f009:**
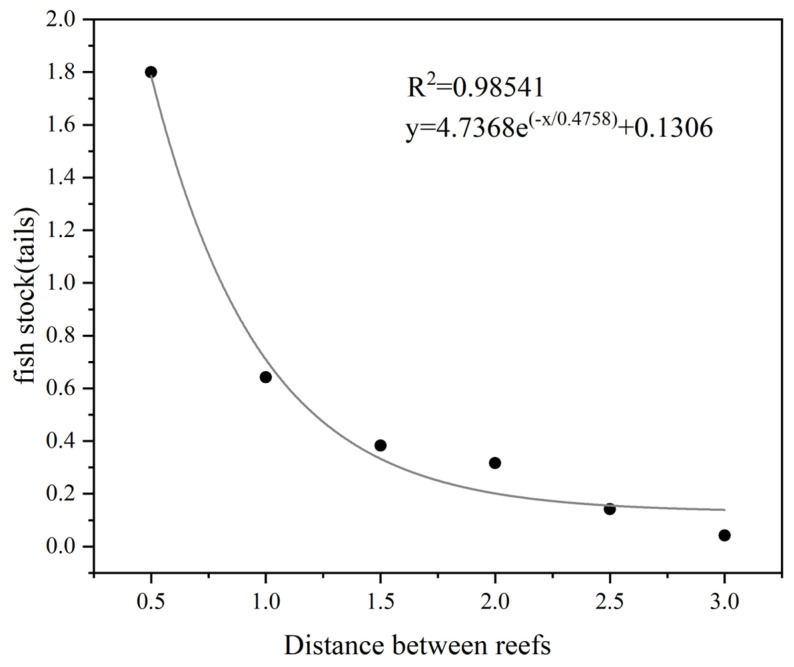
Numbers of fish in Zone A at different reef spacings.

**Figure 10 biology-14-00644-f010:**
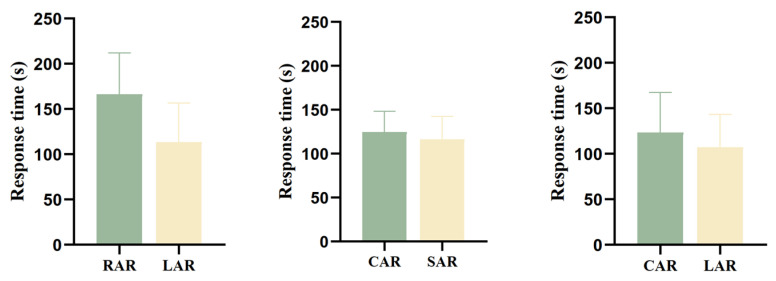
Response times of juvenile *L. erythropterus* in different treatment groups.

**Figure 11 biology-14-00644-f011:**
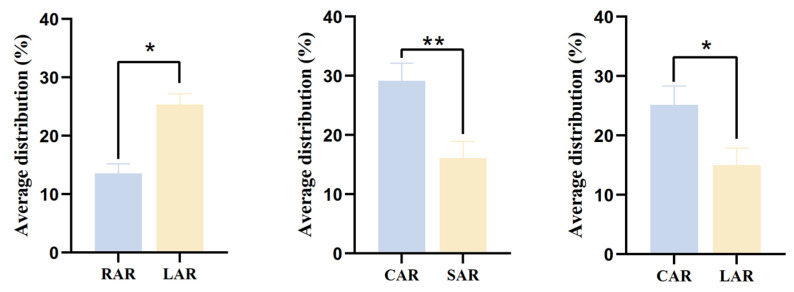
Mean distribution rates of juvenile *L. erythropterus* in different treatment groups. The asterisks in the figure indicate significant correlations between different fish reefs. *: *p* < 0.05; **: *p* < 0.01.

**Figure 12 biology-14-00644-f012:**
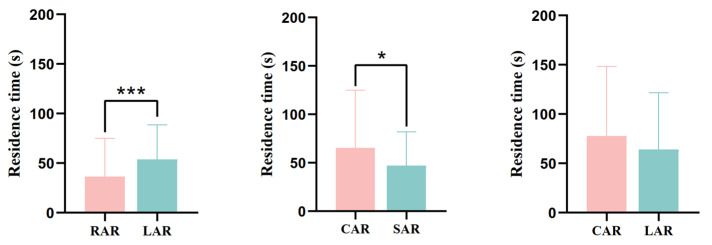
Residence times of juvenile *L. erythropterus* in different treatment groups. The asterisks in the figure indicate significant correlations between different fish reefs. *: *p* < 0.05; ***: *p* < 0.001.

**Figure 13 biology-14-00644-f013:**
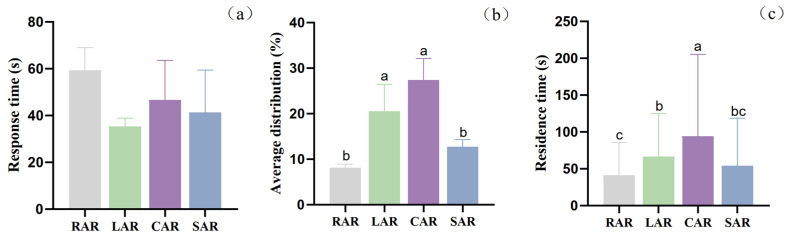
(**a**) Response times of juvenile *L. erythropterus* in different reefs in the presence of four different artificial reef models. (**b**) Mean distribution rates of juvenile *L. erythropterus* in different reefs in the presence of four different artificial reef models. (**c**) Residence times of juvenile *L. erythropterus* in the presence of four different artificial reef models at the same time. Different letters indicate the significant difference at the level of 0.05 between the different fish reef.

## Data Availability

Data are available upon request to the corresponding author.

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
