# Peer review of "Study of the Behavioral Characteristics of Lutjanus erythropterus on the Basis of the Distance and Structure of Artificial Reefs"

_biology, 2025, doi:10.3390/biology14060644_

Round 1
Reviewer 1 Report
Comments and Suggestions for Authors
The study addresses a significant topic in marine ecology and fisheries management, focusing on the design and spacing of artificial reefs to optimize their ecological and economic benefits.
The article can be accepted after the following comments are addressed:
1- What is the meaning of "d" in the spacing terminology? Explain the first instance
2- The introduction is short with limited information. Pls. provide more relevant and recent references to illustrate the previous reports, their achievements, and limitations.
3- Explain the reason for selecting this type of fish. Provide supportive literature reports and/or compare them with similar literature studies.
4- The captions of figures should unified and should appear at the bottom of each figure. Pls, check all the figures accordingly.
5- In conclusion, the authors do not point out possible reasons for the observed behavior. Also, they have not proposed the limitations of the study as well as the future perspectives.
Author Response
Thank you very much for your suggestions and comments on our manuscript. These comments have improved the manuscript effectively. We have included almost all of the suggestions, and below, we present point-by-point responses to the comments.
The article can be accepted after the following comments are addressed:
- What is the meaning of "d" in the spacing terminology? Explain the first instance.
Response to comments: Thank you for your advice. In the manuscript, "d" stands for "distance". We have omitted "times" and expressed "distances of 0.5 times the reef length" as "0.5d". Combined with the opinion of another expert, we changed "d" to "L" (in the revised manuscript, L165).
- The introduction is short with limited information. Pls. provide more relevant and recent references to illustrate the previous reports, their achievements, and limitations.
Response to comments: Thank you for your advice. We have made an overall adjustment to the preface (in the revised manuscript L49–116 and L540–581).
Artificial fish reefs are human-made structures that are constructed and deployed in water areas. By changing the flow structure, these reefs can improve the ecological environment of coastal waters. Moreover, these reefs attract fish, thereby increasing the richness of fishery resources [1, 2]. Compared with natural fish reefs, artificial fish reefs are largely more effective at increasing the diversity of fishery resources and improving fish catch [3-5]. To date, foreign researchers have focused mainly on the distribution and aggregation of fish near artificial fish reefs. Devices such as underwater cameras and echo sounders are usually used to analyze the behavioral patterns of fish [6-8]. In contrast, domestic researchers have focused more on the influences of the design and structure of artificial fish reefs on attraction effects [9, 10]. The attraction effect generated after the deployment of artificial fish reefs is an extremely important reference index in the design of artificial fish reefs. When designing artificial fish reefs, multiple factors need to be comprehensively considered, including the shape of the reef body; the size, quantity, and shape of the openings; and the angle of the flow deflectors [11, 12]. These factors affect the environment surrounding artificial fish reefs and thus influence the attraction effect. However, when artificial fish reefs are constructed for fish habitat restoration, two scientific issues need to be explored: (1) the integrity of the ecological service functions and the structure of the fish reef construction area and (2) the adapt-ability and selectivity of fish behavior in the fish reef construction area. Exploring the behavioral characteristics of reef fish in artificial fish reef environments is helpful for revealing the internal principles of the fish aggregation phenomenon and further im-proving the ecological restoration effect of artificial fish reefs, ultimately enhancing fishery resources.
Reef-dwelling fish are the fish species that mainly inhabit coral reefs, rocky reefs and other reef areas and their surrounding regions. Reef-dwelling fish and their habitats are interdependent. The reefs provide the fish with foraging grounds, breeding sites and safe shelters to avoid natural enemies. Moreover, the material circulation and energy flow of the reef ecosystem cannot do without the participation of the activities of reef-dwelling fish [13, 14]. Artificial fish reefs are important artificial habitats that can improve the marine environment, simulate natural reefs, attract the aggregation of reef-dwelling fish, and create a suitable living space, which is highly important for maintaining population size. Exploring the interaction relationships between artificial fish reefs and fish is highly important for maintaining the stability of marine ecosystems [15].
Lutjanus erythropterus, a representative rocky reef fish, is a common tropical and subtropical fish species. These fish are mainly distributed in the South China Sea, the waters around Taiwan, and the coastal areas of Guangxi, Guangdong, as well as other regions in China [16]. In recent years, owing to the impacts of overfishing and extreme climate events, aquatic ecosystems have been damaged, and the wild resources of the crimson snapper have sharply decreased, posing a serious threat to fishery resources and the ecological balance. Therefore, it has become necessary to understand the behavioral characteristics of this species in its habitat [17].
In recent years, research on artificial fish reefs has emerged continuously, but there are still some deficiencies in certain key areas. In previous studies, scholars often either focused on a single type of artificial fish reef or were limited to relatively closely spaced fish reefs. Relatively little research has been conducted on the comprehensive comparison of various types of fish reefs and the behaviors of fish under a wide range of fish reef spacings. Moreover, the current understanding of the behavioral patterns and preference mechanisms of juvenile L. erythropterus in different reef environments is not comprehensive enough. A reasonable layout of fish reefs can maximize their ecological and economic benefits. Therefore, research on the preferences of juvenile crimson snappers for different types of artificial fish reefs and the interaction distances between fish reefs is particularly urgent and necessary.
Previous research on the behavior of juvenile L. erythropterus has shown that when the distance between fish reefs is greater than three times the size of a single reef, the reefs act as independent entities. However, when the distance is less than 0.5 times the size of a single reef, the reefs form an integrated structure and lose their ability to function as a group of independent fish reefs. Therefore, in this study, by observing the aggregation behaviors of crimson snappers under different fish reef spacings and their behavioral responses when there are two or four different types of fish reefs in the same group (room-shaped artificial reef, RAR; lobe-shaped artificial reef, LAR; cubic artificial reef, CAR; and stacked artificial reef, SAR), a comprehensive analysis of their preference patterns is performed. The aim of this study is to systematically explore the behavioral performance of juvenile crimson snappers in a diverse artificial fish reef environment. This study helps to more accurately understand the ecological needs of juvenile crimson snappers and provides a scientific basis for optimizing the design and layout of artificial fish reefs. The research results not only provide practical references for the selection, configuration, and spacing setting of artificial fish reefs in marine ranches but also positively impact the protection and enhancement of rocky reef fish resources, the maintenance of marine ecological balance, and the promotion of the sustainable development of marine fisheries.
- Explain the reason for selecting this type of fish. Provide supportive literature reports and/or compare them with similar literature studies.
Response to comments: Thank you for your comments. We have supplemented the reasons for choosing Lutjanus erythropterus in the preface section and added references (in the revised manuscript, L81–88).
- The captions of figures should unified and should appear at the bottom of each figure. Pls, check all the figures accordingly.
Response to comments: Thank you for your careful work. This term has been modified in the revised manuscript (L157, L179–180, L325–326, L352, and L363–364).
- In conclusion, the authors do not point out possible reasons for the observed behavior. Also, they have not proposed the limitations of the study as well as the future perspectives.
Response to comments: Thank you for your comments. We have supplemented the behavioral characteristics in the conclusion in the revised manuscript (L516–520). The limitations of the study were added to the discussion in the revised manuscript (L501–507).
Reviewer 2 Report
Comments and Suggestions for Authors
- Need additional experiments: comparison of experiments at distance < 0.5d and distance >3.0d. Or explain why use experimental distances of 0.5d-3.0d.
- There are fewer references in Introduction, please add them.
Author Response
Response to Reviewer Comments
Thank you very much for your suggestions and comments on our manuscript. These comments have improved the manuscript effectively. We have included almost all of the suggestions, and below, we present point-by-point responses to the comments.
Reviewer 2
- Need additional experiments: comparison of experiments at distance < 0.5d and distance >3.0d. Or explain why use experimental distances of 0.5d-3.0d.
Response to comments: Thank you for your comments. Our previous research revealed that, for juvenile Lutjanus erythropterus, when the distance between the artificial reefs in the habitat is less than 0.5 (a certain reference value), the effect of the two reefs is the same as that of a single reef. Therefore, we consider that the two artificial reefs become the same reef body at this time. When the distance between the artificial reefs is more than 3 times the reference value, the two reefs lose their group effect. On this basis, the interaction distance selected in this study is set to a value between 0.5 and 3 times the reference value. We have supplemented the relevant information in the preface of the manuscript (in the revised manuscript, L100–104).
- There are fewer references in Introduction, please add them.
Response to comments: Thank you for your comments. We have made overall revisions to the preface and supplemented the references (in the revised manuscript, L49–116 and L540–581).
Reviewer 3 Report
Comments and Suggestions for Authors
The authors studied the performance of four types of reefs in harboring juvenile Lutjanus erythropterus and the effect of the distance between reefs on the performance of reef units. The results provide important information on selection of reef shape and the arrangement of reef units. Several questions are provided for the authors to consider to improve the quality of this manuscript.
- Suggest the authors using a better title. The title does not describe all the work of this manuscript. Moreover, it is seldom to use the clause (at which) in the title.
- The authors can consider to adjust the order about the study on the effects of the distance, and on the performance of reef type or reef shape. It may be better to present the performance of reef type, and then present the effect of the distance. The cubic reef was proved to be best by studying the performance of the combination of two different reefs, and the combination of four different reefs. Then the cubic reef which had the best performance was selected to study the effect of the distance. If adopting this order, the corresponding changes must be made in the sections of Abstract, Method, results, Discussion and Conclusion.
- When studying the effect of the distance, three different tanks were adopted. Why did not all the tests be conducted in the tank with the length of 550. If is there any effect of the different tanks on the behavior of fish? It’s better to discuss the possible effect of the tanks in the section of Discussion. The authors explained the reason that three different tanks were used at Lines 163-164. But it is difficult to get the meaning.
- The information on the reef type or reef shape is missing in the conclusion.
- Titles of many sections are too long , such as the title of section 2.5.2 and 2.5.3. It is better to refine them.
Other questions and revisions are as follows.
- It seems that it is not appropriate to use ‘d’ to express the distance. Many studies use L.
- delete ‘(CAR)’ at Line 41.
- The meaning of ‘understanding its adaptive and selective behavior’ is not clear at Line 59.
- Add the reference at Line 105.
- The meaning of ‘the preferences of L. erythropterus and fish height (approximately 4.0 cm)’ is not clear at Line 106.
- Replace redfin flute seabream to L. erythropterus at Line 112.
- The meaning of ‘L. erythropterus reef preference’ is not clear between Line 144 and Line 146
- It’s difficult to understand the sentence at Lines 163 to 164.
- Remove Table 1 because the data are the same in Table 1,and Figure 8.
- The maximum value of y tick label needs to be the same for each panel in Figures 10-12. For example, the maximum value of y tick label of the other two panels should be also 250 in Figure 10.
- It seem that the sentences can be removed at Lines 425 to 428.
The meaning of several sentences is not complete or clear. For example, in the sentence 'As a representative rocky reef fish, Lutjanus erythropterus is an economically important fish widely distributed in subtropical and tropical waters, and understanding its adaptive and selective behavior is important for gaining a deeper understanding of the ecological needs of rocky reef fish', the authors might want to express its adaptive and selective behavior related to the habitat.
The similar sentences are listed in the Comments.
Author Response
Response to Reviewer Comments
Thank you very much for your suggestions and comments on our manuscript. These comments have improved the manuscript effectively. We have included almost all of the suggestions, and below, we present point-by-point responses to the comments.
Reviewer 3
The authors studied the performance of four types of reefs in harboring juvenile Lutjanus erythropterus and the effect of the distance between reefs on the performance of reef units. The results provide important information on selection of reef shape and the arrangement of reef units. Several questions are provided for the authors to consider to improve the quality of this manuscript.
- Suggest the authors using a better title. The title does not describe all the work of this manuscript. Moreover, it is seldom to use the clause (at which) in the title.
Response to comments: Thank you for your comments. We have changed the title to “Study of the Behavioral Characteristics of Lutjanus erythropterus on the Basis of the Distance and Structure of Artificial Reefs”.
- The authors can consider to adjust the order about the study on the effects of the distance, and on the performance of reef type or reef shape. It may be better to present the performance of reef type, and then present the effect of the distance. The cubic reef was proved to be best by studying the performance of the combination of two different reefs, and the combination of four different reefs. Then the cubic reef which had the best performance was selected to study the effect of the distance. If adopting this order, the corresponding changes must be made in the sections of Abstract, Method, results, Discussion and Conclusion.
Response to comments: Thank you for your comments. We first investigated the distance effect before studying the preference performance of both the two-reef and four-reef combinations to identify the relatively preferred reef types for Lutjanus erythropterus. This approach was necessary because we needed to consider the placement positions of the reefs. If the distance between reefs is inappropriate, the simultaneous presence of multiple reef types may create combined effects. In such cases, the fish's preference might be attributed to the collective presence of 2, 3, or even 4 reefs together, making it impossible to determine whether they prefer individual reef units or the combined reef configuration. Therefore, prior to conducting the reef-type preference selection experiments, we performed preliminary experiments to determine the distance required for reefs to function independently. This research served as essential foundational work for our subsequent studies.
- When studying the effect of the distance, three different tanks were adopted. Why did not all the tests be conducted in the tank with the length of 550. If is there any effect of the different tanks on the behavior of fish? It’s better to discuss the possible effect of the tanks in the section of Discussion. The authors explained the reason that three different tanks were used at Lines 163-164. But it is difficult to get the meaning.
Response to comments: Thank you for your comments. We employed three different tank sizes on the basis of two key considerations:
- Requirement for experimental synchronization. To minimize the potential impacts of environmental fluctuations (e.g., water temperature, lighting, and water quality) on fish behavior, all the experiments were conducted within similar periods. Using only a single large tank (550 cm) would require the sequential testing of different reef spacings, significantly prolonging the experimental duration and resulting in greater variations in environmental parameters. The adoption of multiple tank sizes enabled synchronous experiments, ensuring that all distance groups were tested under relatively consistent environmental conditions while effectively controlling for temporal variables and improving data comparability.
- Although three different tank sizes were used, we ensured that the distance between each artificial reef and the tank edge was at least twice the reef spacing—a parameter carefully calculated during the experimental design to prevent edge effects. Additionally, using 550-cm tanks for short-distance groups would result in disproportionately small relative distances between reefs, potentially weakening the aggregation effects due to the excessive amount of space and consequently affecting the results.
- The information on the reef type or reef shape is missing in the conclusion.
Response to comments: Thank you for your comments. We have supplemented the behavioral characteristics in the conclusion in the revised manuscript (L516–520).
- Titles of many sections are too long, such as the title of section 2.5.2 and 2.5.3. It is better to refine them.
Response to comments: Thank you for your comments. We have changed the title of sections 2.5.2, 2.5.3, 3.3 and 3.4.
Other questions and revisions are as follows.
- It seems that it is not appropriate to use ‘d’ to express the distance. Many studies use L.
Response to comments: Thank you for your useful suggestions. We have changed "d" to "L".
- delete ‘(CAR)’ at Line 41.
Response to comments: Thank you for your careful work. We have deleted ‘(CAR)’ in the revised manuscript L41.
- The meaning of ‘understanding its adaptive and selective behavior’ is not clear at Line 59.
Response to comments: Thank you for your comments. We have made overall revisions to the preface in the revised manuscript (L49–116).
- Add the reference at Line 105.
Response to comments: Thank you for your comments. We have added this reference to the revised manuscript (L142).
- The meaning of ‘the preferences of L. erythropterus and fish height (approximately 4.0 cm)’ is not clear at Line 106.
Response to comments: Thank you for your careful work. This term has been modified in the revised manuscript (L140–144).
In previous studies, the body height of juvenile L. erythropterus was 4 cm. On the basis of the thigmotactic characteristics of juvenile L. erythropterus toward fish reefs with different opening shapes [1], the opening shape of the fish reef in this experiment was set as circular, and its diameter was set to twice the body height of the juvenile fish (8.0 cm).
- Replace redfin flute seabream to erythropterus at Line 112.
Response to comments: Thank you for your careful work. This term has been modified in the revised manuscript (L149).
- The meaning of ‘ erythropterus reef preference’ is not clear between Line 144 and Line 146
Response to comments: Thank you for your comments. We have reorganized this passage in the revised manuscript (L140–144).
- It’s difficult to understand the sentence at Lines 163 to 164.
Response to comments: Thank you for your comments. This term has been modified in the revised manuscript (L206).
- Remove Table 1 because the data are the same in Table 1, and Figure 8.
Response to comments: Thank you for your careful work. We have deleted Table 1 in the revised manuscript.
- The maximum value of y tick label needs to be the same for each panel in Figures 10-12. For example, the maximum value of y tick label of the other two panels should be also 250 in Figure 10.
Response to comments: Thank you for your useful suggestions. We have changed Figures 10–12 in the revised manuscript.
- It seem that the sentences can be removed at Lines 425 to 428.
Response to comments: Thank you for your comments. We have deleted Lines 425–428.
- The meaning of several sentences is not complete or clear. For example, in the sentence 'As a representative rocky reef fish, Lutjanus erythropterus is an economically important fish widely distributed in subtropical and tropical waters, and understanding its adaptive and selective behavior is important for gaining a deeper understanding of the ecological needs of rocky reef fish', the authors might want to express its adaptive and selective behavior related to the habitat.
Response to comments: Thank you for your advice. We have made overall revisions to the preface and supplemented the references (in the revised manuscript, L49–116 and L540–581). We have contacted AJE to edit the English.
Round 2
Reviewer 1 Report
Comments and Suggestions for Authors
authors have done an excellent revision. The manuscript can be accepted in its current form.
Author Response
Thank you very much for your meticulous review and invaluable feedback on our manuscript.